# Antioxidant, insecticidal activity and chemical profiling of flower's extract of Parthenium weed (*Parthenium hysterophorus* L.)

**Maqsood Ahmed**[1,2], **Ansar Javeed**[3], **Aatika Sikandar**[1], **Mingshan Ji**[1], **Xuejing Bai**[4], **Zumin Gu**[1]*

**1** College of Plant Protection, Shenyang Agricultural University, Shenyang, P.R. China, **2** Department of Agriculture (Plant Protection), Pest Warning and Quality Control of Pesticides, Gujrat, Pakistan, **3** Henan University, Jinming Campus, Kaifeng City, Henan, China, **4** Shenyang Academy of Landscape- Gardening, Shenyang, P.R. China

\* guzumin1212@163.com

**Data Availability Statement:** All relevant data are within the paper.

**Funding:** This research was supported by National Key Research & Development Program of China

## Abstract

*Parthenium hysterophorus* L., an invasive alien species and notorious weed, offers various benefits to the medical and agrochemical industries. This study aimed to evaluate the antioxidant and insecticidal activities of *P. hysterophorus* flower extract and conduct chemical profiling to identify the phytoconstituents responsible for these biological effects. The antioxidant activity was assessed using the 1,1-diphenyl-2-picrylhydrazyl (DPPH) assay, while gas chromatography mass spectrometry (GCMS) analysis was employed for chemical configuration evaluation. Our findings demonstrate that the dichloromethane (DCM) extract of *P. hysterophorus* exhibits potent radical scavenging activity (95.03%). Additionally, phytochemical analysis revealed significant amounts of phenols and flavonoids in the distilled water and ethyl acetate extracts (103.30 GAEg$^{-1}$ and 138.67 QEg$^{-1}$, respectively). In terms of insecticidal activity, the flower extract displayed maximum mortality rates of 63.33% and 46.67% after 96 hours of exposure at concentrations of 1000 µgmL$^{-1}$ and 800 µgmL$^{-1}$, respectively, with similar trends observed at 72 hours. Furthermore, the *P. hysterophorus* extracts exhibited LC$_{50}$ values of 1446 µgmL$^{-1}$ at 72 hours and 750 µgmL$^{-1}$ at 96 hours. Imidacloprid, the positive control, demonstrated higher mortality rates at 96 hours (97.67%) and 72 hours (91.82%). Moreover, the antioxidant activity of *P. hysterophorus* extracts exhibited a strong correlation with phenols, flavonoids, and extract yield. GCMS analysis identified 13 chemical compounds, accounting for 99.99% of the whole extract. Ethanol extraction yielded the highest percentage of extract (4.34%), followed by distilled water (3.22%), ethyl acetate (3.17%), and dichloromethane (2.39%). The flower extract of *P. hysterophorus* demonstrated significant antioxidant and insecticidal activities, accompanied by the presence of valuable chemical compounds responsible for these biological effects, making it a promising alternative to synthetic agents. These findings provide a novel and fundamental basis for further exploration in purifying the chemical compounds for their biological activities.

(Grant No. 0201300). The funders had no role in study design, data collection and analysis, decision to publish, or preparation of the manuscript.

**Competing interests:** The authors declare no conflict of interest.

## Introduction

The introduction of invasive alien species, such as *Parthenium hysterophorus*, has devastating consequences for native biota and ecosystems, contributing to climate change, pollution, and habitat destruction. Understanding the positive and negative impacts of this species can aid in controlling biological invasions, which pose a global risk to biodiversity and agricultural production. Invasive weeds not only compete with crops for resources but also produce bioactive compounds that hold potential for medicinal purposes, highlighting the importance of natural plants in pharmaceutical and herbal systems [1].

*Parthenium hysterophorus* L. is an extremely invasive plant species belonging to family Asteraceae that has invaded almost fifty countries in sub-continent Africa, Asia, and Oceania, infesting forests, feeding lands, and urban landscapes [2, 3]. It is a devastating and hazardous weed of various economically important crops [4]. Moreover, it is amongst the top ten worst weeds in the world and is included in the world catalogue of destructive species [5, 6]. In Europe *P. hysterophorus* is considered as an ephemeral species. It is generally recognized by different names such as parthenium weed, congress grass, Gajjar botti, white head and white top [7]. It is native to Gulf of Mexico and United States. However, it accidently introduced in Indian subcontinent [8]. This weed has been considered as a major weed of high rank spreading at a much faster pace to takeover abandoned land, canal, watercourse banks, roads, and fields to challenge crop productivity [9, 10].

Medicinal plants are the fundamentals of traditional therapeutic systems based on the trusts and observations comprised of hundreds of years [11]. The phytochemical constituents such as phenolic and flavonoid are secondary metabolites involved in therapeutic system, because of the existence of these functional phytochemicals mainly parthenin [12]. Moreover, *P. hysterophorus* renowned to contain vital biochemical compounds such as alkaloids, tannins, and saponins, which are important from the medicinal viewpoint [13]. It also exhibited allelopathic effects and important phytochemicals such as parthenon and coronopilin which are auto-toxic to its own seed germination and seedling growth [14]. Furthermore, serious health consequences on human beings and livestock have been reported [15]. The agriculturists who are come in contact with *P. hysterophorus*; the pollen and dust produced by this weed provoke allergic contact dermatitis in human beings. Major chemical compounds, parthenin's contained by *P. hysterophorus* exhibited phytotoxic properties against a variety of plants, including weeds and various crops such as Slender amaranth (*Amaranthus viridis*), Billygoat weed (*Ageratum conyzoides*), common wild oat. (*Avena fatua*), nettleleaf goosefoot (*Chenopodium murale*), Foetid cassia (*Cassia tora*), mung bean (*Phaseolus aureus*) [16, 17]. Moreover, parthenin's the main biochemical compounds showed insecticidal activity against termites, cockroaches and migratory grasshoppers, *Melanoplus sanguinipes* [7]. Similarly, Patel and Chitra [18] reported pesticidal activity of the ethanol extract of *P. hysterophorus* which showed substantial lethal effect against stored grain pest like *Tribolium* sp. and *Oryzaephilus* sp. Essential oils from the natural plants also have antioxidant effects. In recent years several reports have been breakout, the essential oil from Rosemary '(*Salvia rosmarinus*) [19], wild mint (*Mentha longifolia* var. *Calliantha*) [20], black turmeric (*Curcuma caesia*) [21] *Ammodaucus leucotrichus* [22] and black pepper (*Piper nigrum*) [23].

Pharmacologically, it is involved in treating numerous diseases like allergies, anemia, sores, skin diseases, facial neuralgia, fever, vermifuge, blood purifier, and insecticide [7, 24]. This plant species is enriched with vital phytoconstituents however; sesquiterpene lactones are the most copious phytochemicals exhibited by this plant which are accountable for injurious effects such as strong allelopathic [25], toxic to livestock [26], It also possess many advantageous effects like antibacterial [27, 28], acaricidal [29], antioxidant [30, 31], cytotoxic [32, 33],

larvicidal potential against *Aedes aegypti* L. [34, 35], skeletal muscle relaxant [36], pesticidal agent [37], producing bioethanol [38], as reducing agent in the formation of silver nanoparticles [39] and in production of zinc nanoparticles [40]. However, various factors affect the biological profile of the plants and from some recent studies it has been reported that nanoparticle influenced the physiological and biological properties of the plants. It has been reported that metal-based nanoparticle showed a decrease in protein and amino acid and micronutrient contents in wheat crop grown under such conditions [41]. Moreover, nitrogen doped titanium dioxide nanoparticle enhanced antioxidant activity via reducing power assay and percent inhibition of DPPH against *Streptomyces* [42]. Likewise, synthetically and biologically prepared nanoparticle significantly exerted effects on plant growth and secondary metabolites which ultimately triggered the biological activity of the plants [43].

As some earlier researches have been conducted on this plant species but, detailed studies on chemical and quantitative phytochemical analysis and assessment of antioxidant activities from different solvents extract as well as aphicidal activity against cabbage aphid *Brevicoryne brassicae* from flower's extracts has not been explored before. Hence, keeping in view the diverse biological properties of *P. hysterophorus*, current study was designed to determine the antioxidant and insecticidal activity of flower's extract against *B. brassicae* with appraisal of quantitative phytochemical analysis as well as assessment of chemical profiling of extract.

## Materials and methods

### Collection and samples preparation

The whole plant samples were collected from District Sialkot Punjab Pakistan during the month of June-July 2019 and were identified as *Parthenium hysterophorus* L., from Ayub Agricultural Research Institute (AARI) Faisalabad Pakistan by Dr. Dilbar Hussain. Flowers were manually separated from the plants and left over for 20 days to dry under shade. These dried flowers were powdered by electric grinder and then extracted with different solvents (ethanol, ethyl acetate, dichloromethane and distilled water) by cold extraction method for seventy two hours at fixed temperature 28°C and at 100 rpm in an incubator shaker (ZWY-1102C). Extracted contents were filtered and concentrated to lessen the volume on rotary evaporator model R-210 BUCHI Labortechnik AG, CH-9230 Flawil 1/Switzerland). Obtained extract's yield was measured using following Equation (Eq 1).

$$\text{yield } (\%) = \frac{(\text{Weight of the extract})}{(\text{Weight of the dried sample})} \times 100 \tag{1}$$

### Quantitative analysis for total phenolic and flavonoids content

To assess the phytochemicals quantitatively, 1mg of extract from each solvent was dissolved in 1mL Methanol separately. Consequent mixture was vortexed and separated for analysis. For the assessment of total content for phenols Folin—Ciocalteu reagent test was used. In brief, 1mL solution (1mgmL$^{-1}$) was added to 2.5mL of (10%) Folin—Ciocalteu with supplementary addition of 2mL solution of (2%) sodium carbonate ($Na_2CO_3$). The consequential blend was incubated for fifteen min in the dark to measure absorbance at 765nm in 96-well ELISA plate using (SpectraMax 190, Meigu molecular International Co. Ltd., Shanghai China). To build a standard curve, Gallic acid was used (1mgmL$^{-1}$) at different concentrations such as 1, 0.50, 0.25, 0.10, 0.05, 0.02, 0.01, 0 mgmL$^{-1}$ to describe the obtained results as Gallic acid equivalent (GAE) mgg$^{-1}$ of total extract. Each treatment was replicated ten times to analyze the data.

To examine the total flavonoids content, aluminium chloride colourimetric method was adopted. Briefly, 1mL solution (1mgmL$^{-1}$) was poured into 3mL methanol, with addition of 0.2mL of 1M Potassium acetate (CH$_3$COOK), 0.2mL of (10%) aluminium chloride (AlCl$_3$) and at the end, distilled water 5.6mL was added to the consequent mixture. The incubation of solution was performed for half an hour in the darkness and absorbance at 420nm was measured. A standard curve was built using quercetin at different concentrations such as 1, 0.50, 0.25, 0.10, 0.05, 0.02, 0.01, 0 mgmL$^{-1}$ to represent the results as quercetin equivalent (QE) mgg$^{-1}$ of total extract. The procedure was replicated 10 times for each treatment.

### Assessment of free radical scavenging activity

Antioxidant activity of extracts obtained by different solvents from *Parthenium hysterophorus* flowers was accomplished in 1% solution of tween 20 by using 1,1-diphenyl-2-picrylhydrazyl (DPPH) (C$_{18}$H$_{13}$N$_5$O$_6$) a free radical. In brief, to assess the free radical scavenging activity, all dried solvent extracts were mixed together in equal proportion (5mg each) and were dissolved in HPLC grade methanol and filtered. The filtrate was formerly used to assess the antioxidant activity. Similarly, DPPH solution was prepared in same grade Methanol @ 0.001g 25mL$^{-1}$. Then, into 3.5mL freshly prepared DPPH solution and 0.25mL extract prepared in Methanol was added. The consequent mixture was shacked for complete mixing and then the resulting mixture was incubated in darkness at 28°C for 30 min [44]. After incubation of resulting mixture for stipulated period of time, the absorbance was assessed at 517nm by using (SpectraMax 190, Meigu molecular International Co. Ltd., Shanghai China). The inhibition percentage of primed DPPH solution was predicted on reduction of absorbance through following Equation (Eq 2). Each treatment was replicated 10 to analyze the data.

$$\text{Inhibition } (\%) = \frac{A_{blank} - A_{sample}}{A_{blank}} \times 100 \qquad (2)$$

Where: A$_{blank}$ = (control absorbance); A$_{Sample}$ = (samples absorbance).

### Correlation of antioxidant activity versus phenols, flavonoids and extract yield

Correlation of antioxidant activity *P. hysterophorus* versus Phenols, Flavonoids and extract yield was carried out using SPSS statistics 25.0 version with significant values at (P≤0.05) level.

### Assessment of insecticidal activity

Residual toxicity method was adopted for the assessment of insecticidal activity of extract of *P. hysterophorus* flowers. Samples for bioassay study were prepared to obtain mother mixture by mixing 0.1g of each solvent's extract and dissolved in Acetone, because it is aprotic solvent having intermediate polarity comparable to polar parotic solvents and it also evaporates immediately from the extract when exposed to air. The resulting mixture was placed in the fume hood for six hours for complete drying of Acetone. Serial concentrations *viz*. 100, 200, 400, 800 and 1000 μgmL$^{-1}$ was prepared in Tween-20 (1%) solution from mother mixture. Briefly, fresh cabbage leaf discs 5cm in diameter were cut off and dipped for 10 s in the respective concentration and then dried for 5 min at room temperature. Formerly, 10 adult wingless aphids were carefully released onto the leaf disc using a fine-haired brush contained in the Petri dish and incubated for 96 h at 65% R.H., 25°C and with a 16:8 (light: dark) photoperiod. Imidacloprid 25% WP was used as the positive control at a rate of 0.0025 mLmL$^{-1}$ of water, and control (CK) with a 1% Tween 20 solution. Positive controls and CKs were placed in a greenhouse for

96 h. Mortality data was collected at 12, 24, 48, 72 and 96h using binocular microscope and the response of aphids was observed by needle probing, and the aphids who offered no response on stimulation were considered dead. Each treatment was replicated five times.

### Chemical analysis

Samples for analysis were prepared by mixing 0.1g of each solvent extract into 2mL of HPLC grade methanol to obtain mother solution. Consequent mixture was vortexed for 1 min following the addition of graphitized carbon black "GCB" (0.005g). Mixture was centrifuged for 10 min at 10000 rpm at 25˚C and colorless material was separated and collected in (1.5mL) centrifuges tubes. Work for GCMS analysis was accomplished using an Agilent Model 6890-5973N accompanied with gas chromatograph established on HP1 capillary column TG-5MS polydimethylsiloxane (length 30m × diameter 250μm × film thickness 0.25μm) interfaced with mass selective detector Hewlett Packard (5973N). The established parameters were; preliminary temperature was adjusted at 70˚C for 0 min however, final temperature was extended to 200˚C for 10min$^{-1}$ whereas, inlet temperature was adjusted at 250˚C with split ratio 10:1. MS quadruple temperatures were fixed at 150˚C while thermal aux temperatures were 285˚C. The Scan range for MS was 35–520 units however; as a carrier, helium gas was used bearing stream frequency of 1.0 mLmin$^{-1}$. The existing compounds in the extract were identified by comparison of their retention time and the mass spectra of the NIST 08 and Wiley 7 of the data base and confirmed by comparing with GCMS literature data at NIST and TMS database Wiley/NIST.1998.1 [45].

### Statistical analysis

All calculated data was analysed by one way analysis of variance (ANOVA), mean difference between treatments was calculated for significance test by Duncan Multiple Range Test "DMRT" at $P \leq 0.05$) with IBM-SPSS statistics 25.0 version software. Probit analysis was performed using EPA Probit analysis program version 1.5.

## Results

The current study carried out on the flower's extract of *Parthenium hysterophorus* revealed the existence of medicinally and agro-chemically active compounds. The phytochemical constituents of *P. hysterophorus* were qualitatively analyzed for antioxidant and insecticidal activities.

### Extract yield (%)

Yield of extract by the solvent extraction technique from *P. hysterophorus* flowers was calculated and presented in (Fig 1).

The maximum extract was obtained by ethanol (4.34%) whereas; minimum extract quantity (2.39%) was afforded by dichloromethane. The extract obtained via ethanol and ethyl acetate was bluish green and greenish in color and exhibited consistency, respectively. The same oil was brownish in color and non-consistent via dichloromethane and distilled water, respectively, (Table 1).

### Analysis for phenol and flavonoid and antioxidant activity

The total contents for phenol and flavonoid as well as DPPH scavenging activity afforded by *P. hysterophorus* flower's extract via different solvents are presented in (Table 2). Results confirmed that the highest total phenol were reported from distilled water 103.30 GAE/g followed by Ethyl acetate and Ethanol' extract, 47.79 and 27.55 GAEg$^{-1}$, respectively. On the other hand

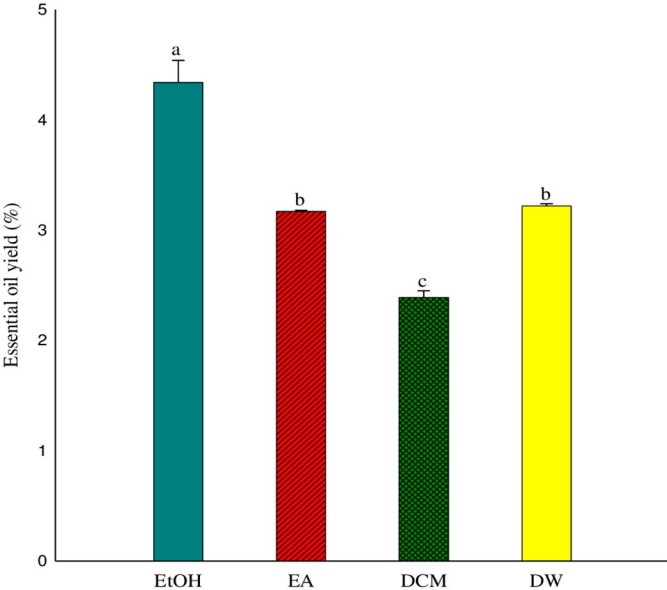

**Fig 1. Yield of extract obtained by different solvents: Data presented in bars is described as mean values ± standard error with superscripts is significantly different according to "DMRT" $P \geq 0.05$).** EtOH; (Ethanol), EA; (Ethyl Acetate), DCM; (Dichloromethane), DW; (Distilled Water).

highest flavonoid was reported from Ethyl acetate extract 138.67 QEg$^{-1}$ followed by Ethanol 14.23 QEg$^{-1}$ respectively.

Inhibition (%) was determined by 1,1-Diphenyl-1-picrylhydrazyl (DPPH) displayed color absorption at 517nm on using spectrophotometer. When DPPH trap free radicals, the color of the solution changed to light yellow and, subsequently, resulted in decrease of the absorbance. Maximum DPPH radical scavenging activity was afforded by Dichloromethane extract 95.03% followed by Ethanol and Ethyl acetate 88.53% and 67.21%, respectively.

## Correlation of antioxidant activity versus phenols, flavonoids and extract yield

In our study, the Pearson's correlation regarding antioxidant activity of *P. hysterophorus* showed positive relationship (Table 3). Results showed highly significant ($P \leq 0.01$) negative effect of phenol with DPPH as well as DPPH showed strongly highly significant negatively correlation and similar results were found reciprocally in the studied parameters. However, yield extracted by different solvents did not affect the Antioxidant Activity versus phenols, flavonoids quantitatively.

**Table 1. Physical appearance of the flower's extract of *Parthenium hysterophorus*.**

| Extract | Visibility | Physical Nature | Consistency |
|---------|-----------|-----------------|-------------|
| EtOH | Bluish green | Oily | Consistent |
| EA | Greenish | Oily | Consistent |
| DCM | Brownish | Oily/Gummy | Non-consistent |
| DW | Brownish | Oily/Gummy | Non-consistent |

EtOH; (Ethanol), EA; (Ethyl Acetate), DCM; (Dichloromethane), DW; (Distilled water).

**Table 2. Total phenol, flavonoid content and DPPH radical scavenging activity of the flower's extract of *Parthenium hysterophorus*.**

| Extracts | Phenol (GAEg$^{-1}$) | Flavonoid (QEg$^{-1}$) | DPPH Inhibition (%) |
|---|---|---|---|
| EtOH | 27.55±1.46[c] | 14.23±0.10[b] | 88.53±0.11[b] |
| EA | 47.79±0.85[b] | 138.67±2.20[a] | 67.21±0.19[c] |
| DCM | 30.50±0.00[d] | 15.38±0.04[c] | 95.03±1.58[a] |
| DW | 103.30±1.16[a] | 4.81±0.01[c] | 18.73±0.89[d] |
| Statics | S.S = 24268 | S.S = 248063 | S.S = 13880 |
| | M.S = 8069 | M.S = 82687 | M.S = 4626 |
| | Df = 3 | Df = 3 | M.S = 3 |
| | $f$ = 2564*** | $f$ = 22488*** | $f$ = 1031*** |

Data in the columns is described as mean values ± standard deviations with various superscripts are significantly different according to DMRT" $P \leq 0.05$). S.S (Sum of square); M.S (Mean square); Df (Degree of freedom); $f$ (Significance);

*** (significance level).

## Assessment of insecticidal activity

The mortality data of cabbage aphid *B. brassicae* at specific period of time using *P. hysterophorus* flower's extract is presented in (Fig 2).

It was perceived that the percent mortality was directly correlated to the concentration and the time. The results demonstrated that the maximum mortality was recorded at 1000μgmL$^{-1}$ at 96 h and 72 h with 63.33% and 46.67%, respectively. Same trend of mortality was also observed at 800 μgmL$^{-1}$ with 56.67% and 36.67%, respectively. Imidacloprid 25% WP being a positive control produced the highest rates of mortality after 96 h of exposure 97.67%. Likewise, Imidacloprid also produced significant mortality after 72 h and 48 h exposure 91.82% and 87.38% mortality, respectively.

Analysis for probability of the data showed the LC$_{50}$ values, slope, chi-square, and fiducial limits at confidence interval of 95% for *P. hysterophorus* extracts. Mortality response using plant extracts showed sensitivity of *B. brassicae* to various concentrations which is represented in (Table 4).

## Chemical analysis

The existence of chemical compounds in *P. hysterophorus* flower's extract was analyzed by GCMS techniques and data is presented in (Table 5). Thirteen biochemical compounds equivalent to 99.99% of entire extract by GC fraction were identified. Moreover,2-naphthalene-methanol,decahydro-.alpha.,.alpha.,4a-trimethyl-8- methylene-, [2R—(2.alpha., 4a.alpha.,8a. beta.)]-(25.93%), histidine, 1,N-dimethyl-4-nitro- (19.25%) and 3-methoxy-3, 4-dimethyl-

**Table 3. Correlation of the antioxidant activity of *P. hysterophorus* extract versus phenols, flavonoids and extract yield.**

| Treatments | Phenols | Flavonoids | DPPH | Yield |
|---|---|---|---|---|
| Phenols | 1 | -0.16 | -0.99** | -0.11 |
| Flavonoids | -0.16 | 1 | -0.71 | -0.90 |
| DPPH | -0.99** | -0.71 | 1 | -0.10 |
| Yield | -0.11 | -0.09 | -0.10 | 1 |

Note:

* Correlation is significant at 0.05 levels;

** Correlation is highly significant at 0.05 levels

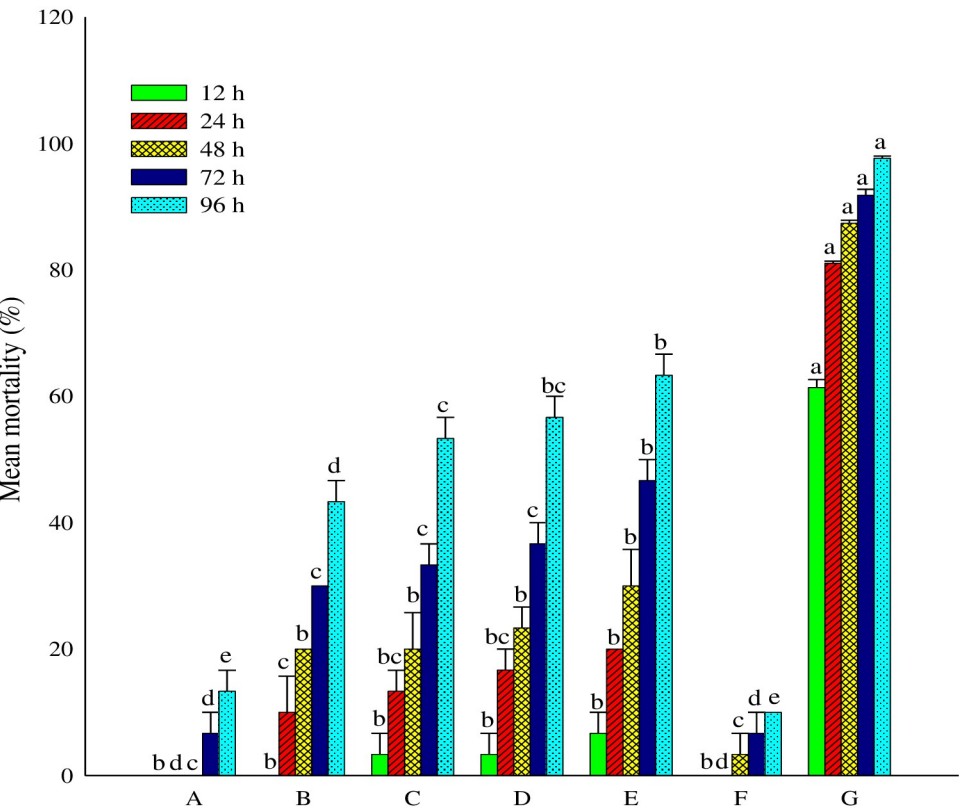

**Fig 2. Insecticidal activity of extract of *Parthenium hysterophorus* flowers: Data in bars is described as mean values ± standard error with various superscripts is significantly different according to "DMRT" *P* ≥0.05).** A (100μgmL$^{-1}$); B (200μgmL$^{-1}$); C (400μgmL$^{-1}$); D (800μgmL$^{-1}$); E (1000μgmL$^{-1}$); F (-iv Control); G (+ iv Control).

1-heptyne (11.27%), were the major chemical compounds while, 10 others chemical compounds were identified in minor amounts having areas with peak range 2.04–3.7%. GCMS chromatogram of *P. hysterophorus* flower extract is presented in (Fig 3).

## Discussion

*Parthenium hysterophorus*, known as Parthenium weed, is recognized as a noxious weed with detrimental effects on ecosystems. However, our study revealed that this invasive plant species possesses valuable properties in terms of antioxidant and insecticidal activities.

**Table 4. Toxicity of *P. hysterophorus* extracts against *B. brassicae* at 12, 24, 48, 72 and 96 hours exposure.**

| Time | LC$_{50}$ μgmL$^{-1}$ | 95% F.L | | Slope± S.E | χ2 |
|---|---|---|---|---|---|
| | | **Lower** | **Upper** | | |
| 12 | 3598 | 1421 | 35512 | 1.09±0.38 | 0.25 |
| 24 | 2569 | 1321 | 27758 | 1.51±0.46 | 0.99 |
| 48 | 2345 | 1153 | 25517 | 1.31±0.53 | 0.85 |
| 72 | 1446 | 852 | 21721 | 1.56±0.63 | 0.22 |
| 96 | 750 | 468 | 1276 | 1.89±0.58 | 0.04 |

Note: LC$_{50}$ (Lethal concentration); S.E. (standard error); χ2 (chi-square); F.L. (Fiducial limit)

**Table 5. Chemical composition of the flower's extract of *Parthenium hysterophorus*.**

| Peak # | R. T | Area % | Chemical Compounds | M.F | M.W gmol$^{-1}$ |
|---|---|---|---|---|---|
| 1 | 3.22 | 7.05 | Benzene, 1,3-dimethyl- | $C_8H_{10}$ | 106.17 |
| 2 | 3.92 | 3.47 | 2,4-Dimethylamphetamine | $C_{11}H_{17}N$ | 163.26 |
| 3 | 4.43 | 2.04 | 3-Piperidinol | $C_5H_{11}NO$ | 101.15 |
| 4 | 5.05 | 2.74 | 2-Pentene, 4,4-dimethyl-, (E)- | $C_7H_{14}$ | 98.186 |
| 5 | 5.52 | 3.55 | 2-Butanamine | $C_6H_5N$ | 101.19 |
| 6 | 12.96 | 25.93 | 2-Naphthalenemethanol,decahydro.alpha.,.α.,4a-trimethyl-8-methylene-,[2R (2 α.,4a.α.,8a.beta.)]- | $C_{15}H_{26}O$ | 222.36 |
| 7 | 14.29 | 11.27 | 3-Methoxy-3,4-dimethyl-1-heptyne | $C_{10}H_{18}O$ | 154.25 |
| 8 | 15.21 | 4.76 | Undecane | $C_{11}H_{24}$ | 156.31 |
| 9 | 15.85 | 3.82 | 2-Pyrrolidinone, 1-methyl- | $C_5H_9NO$ | 99.13 |
| 10 | 16.62 | 7.26 | Glycinamide hydrochloride | $C_2H_7C_lN_2O$ | 110.54 |
| 11 | 17.58 | 2.18 | 1,2-Benzenedicarboxylic acid, butyl 2-ethylhexyl ester | $C_{20}H_{30}O_4$ | 334.45 |
| 12 | 18.1 | 6.67 | Hexadecanoic acid, ethyl ester | $C_{18}H_{36}O_2$ | 284.48 |
| 13 | 21.85 | 19.25 | Histidine, 1,N-dimethyl-4-nitro- | $C_8H_{10}N_2O_2$ | 166.18 |

R.T (Retention time); M.F (Molecular formula); M.W (Molecular Weight).

The chemical compounds found in *P. hysterophorus*, such as phenolics, steroids, alkaloids, flavonoids, and terpenoids, vary in concentration based on ecological factors and pathogen presence. These compounds exhibit bio-pesticidal properties, specifically targeting insect pests and pathogens. Importantly, they are biodegradable and do not pose significant harm to the environment [27, 46–48].

In addition to its antioxidant activity, the flower extract of *P. hysterophorus* exhibited promising insecticidal activity against *Brevicoryne brassicae*, a notorious sucking pest. This finding suggests that the extract has the potential to be used as a natural insecticide. The observed LC$_{50}$ values indicate the concentration at which the extract is lethal to 50% of the exposed insects. The lower LC$_{50}$ value at 96 hours compared to 72 hours indicates the increased effectiveness of the extract over time. These results are consistent with previous studies that have reported the

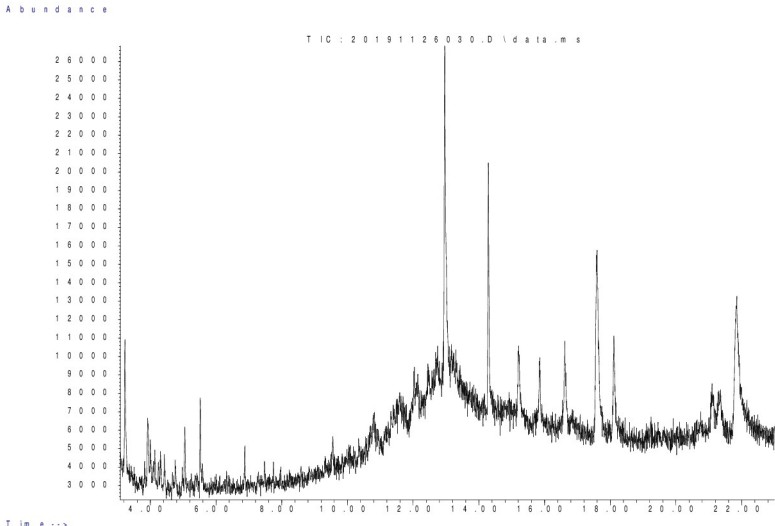

**Fig 3. GC-MS chromatogram of *Parthenium hysterophorus* flower's extract.**

insecticidal properties of plant extracts against various pests [49]. The assessment of antioxidant and polyphenolic activity in medicinal products is crucial for understanding the therapeutic potential of natural plants. Kumar and Mishra [44] identified alkaloids, flavonoids, terpenoids, and cardiac glycosides in the extract of *P. hysterophorus*, while the flower extracts exhibited phenolic contents ranging from 86.69–320.17mg propyl gallate equivalent $PGEg^{-1}$. These findings align with our study, highlighting the significant antioxidant activity of the bioactive compounds present in *P. hysterophorus* extracts. Previous research on *P. hysterophorus* leaves extract reported a 50% radical scavenging potential at $250\mu gmL^{-1}$ in the DPPH assay and 30% inhibition at $100\mu gmL^{-1}$ in hydrogen peroxide assays [50], indicating the antioxidant potential of the plant extract. This antioxidant activity in the hydro-ethanolic extracts of *P. hysterophorus* leaves is likely attributed to the presence of polyphenolic compounds [11, 51]. The flower extract of *P. hysterophorus* exhibited significant antioxidant activity, as evidenced by its potent radical scavenging ability. This finding is in line with previous studies that have reported the antioxidant potential of various plant extracts [52]. The presence of phenolic and flavonoid compounds in the extract may contribute to its antioxidant activity. Phenols and flavonoids are well-known antioxidant compounds that can effectively neutralize free radicals and protect against oxidative stress [53]. The strong correlation observed between antioxidant activity, phenolic/flavonoid content, and extract yield further supports the potential of *P. hysterophorus* as a source of natural antioxidants.

Our findings on the antioxidant activity of *P. hysterophorus* are consistent with previous studies. Sinha and Paul [54] reported similar scavenging properties (54.5%) and $IC_{50}$ values (60.2) via the DPPH assay, as well as high scavenging activity (93.2%) and $IC_{50}$ values (33.6) via the ABTS assay. Likewise, Bashir [55] demonstrated the presence of significant phenolic and flavonoid contents (105.44mg $GAEg^{-1}$ and 41.50mg $REg^{-1}$, respectively) in a 60% ethanolic leaf extract of *P. hysterophorus*, along with comparable $IC_{50}$ values for DPPH radical scavenging activity (87.55 $\mu gmL^{-1}$) and ABTS (98.22 $\mu gmL^{-1}$). Furthermore, the phytochemical profiling of *P. hysterophorus* revealed the presence of terpenes, fatty acids, hydrocarbons, and phytosterols, with the root extract exhibiting the highest antioxidant potential [56]. These findings align with Sinha and Paul [54], who reported substantial antioxidant activity in crude leaf extracts of *P. hysterophorus*, with $IC_{50}$ values of 60.24% and 33.60% using the DPPH and ABTS methods, respectively.

Previous research has demonstrated that exposure of natural fauna to harmful chemicals can have significant impacts on biological activities. For instance, estrogen compounds have been found to cause acute toxicity in edible crops and vegetables, particularly when long-term irrigation with estrogen-containing water is involved. Adeel and Zain [57] reported similar results, showing the uptake and accumulation of l7β-estradiol (17β-E2) and ethinylestradiol (EE2) in lettuce (*Lactuca sativa*) grown under laboratory conditions. These findings emphasize the potential risks associated with the contamination of agricultural systems by estrogen compounds.

The utilization of secondary metabolites, including terpenoids, alkaloids, and flavonoids, as pest control agents has gained attention due to historical practices involving nicotine and pyrethrum [58]. Our findings align with previous studies by Tesfu and Emana [59], who observed significant mortality rates of *Callosobruchus chinensis* after 96 hours of exposure to different parts of *P. hysterophorus* extract. Inflorescence and stem powder demonstrated mortality rates of 76.67% and 56.67%, respectively, while the leaves extract exhibited repellent effects of 70–100% on *Aedes aegypti* adults. Additionally, the diethyl ether extracts showed exceptional repellency at 99.7%.The insecticidal effects of plant extracts can be attributed to their repellency and stomach poisoning properties, which hinder insect movement and cause death by constricting their spiracles [60]. Chemical analysis of *P. hysterophorus* revealed the presence of

sesquiterpene lactones, particularly parthenin derivatives, which exhibited insecticidal and nematicidal activities against *Callosobruchus maculatus* adults and *Meloidogyne incognita* root knot nematodes. Notably, the pyrazoline compound demonstrated high efficacy as an insecticide, with an $LC_{50}$ value of 32mgL$^{-1}$ after 72 hours of exposure, and also exhibited strong nematicidal activity, with an $LC_{50}$ value of 512mgL$^{-1}$ for parthenin after 72 hours [37]. The chemical profiling of the *P. hysterophorus* flower extract using GCMS analysis revealed the presence of several chemical compounds, indicating its complex chemical composition. These compounds could be responsible for the observed biological activities, including antioxidant and insecticidal effects. Further investigation is warranted to isolate and purify these bioactive compounds and evaluate their specific biological properties. The identification and characterization of these compounds can contribute to the development of natural products for antioxidant and insecticidal purposes.

P. hysterophorus extract exhibited remarkable effectiveness against *Lipaphis erysimi*, reducing infestation by 29%, possibly due to the presence of phenolic acids [61]. Ahmad and Bagheri [56] reported mortality rates of 26.70% and 27% in mustard aphids treated with *P. hysterophorus* extract concentrations of 10% and 20% respectively, after 21 days. Additionally, Wasu and Yogesh [62] demonstrated high mortality (100% with freshly prepared solution, 80% with stored stock) of *Aedes aegypti* larvae when exposed to *P. hysterophorus* leaf and stem extract (200g500$^{-1}$ml). Another study highlighted significant mortality rates of 73.33%, 86.67%, and 83.33% against *B. brassicae* and moderate mortality rates of 26.67%, 46.67%, and 63.33% against *Myzus persicae* at 10% concentration and exposure times of 24, 48, and 72 hours respectively [58]. Furthermore, Wasu and Yogesh [62] reported promising effects of crude extracts from fresh *P. hysterophorus* leaves against *Spodoptera littura* and *Spodoptera littorallis*, with $LC_{50}$ values of 7.96% and 8.14% respectively.

Gas chromatography-mass spectrometry (GCMS) analysis revealed the presence of various phytochemicals in different parts of *P. hysterophorus*. Wang and Gan [63] identified fourteen chemical compounds, including phytosterols, stigmasterol, daucosterol, dtigmasterol-3-O-glucopyranoside, and Stigmasteryl-3β-arachidate. Similarly, Białoń et al [64] found diverse terpenoids in Lavender oil, exhibiting antibacterial and antifungal properties. The chemical composition of *Satureja montana* and its tincture demonstrated high polyphenol, phenolic, and flavonoid contents, suggesting potential antimicrobial and antioxidant activities for further clinical investigation [65]. GCMS coupled with the NIST library enabled the identification of known compounds in extracts, facilitating the separation and exploration of unidentified compounds. In another study, phenolic compounds from *Cladofora glomerata* algae were isolated and characterized using colorimetric, chromatographic, and HPLC methods, revealing the presence of nine phenolic compounds, phenolic acids, and flavonoids [66]. In the leaf extracts of *P. hysterophorus*, a significant abundance of parthenin, a key component, was observed, showing a positive correlation with cytotoxicity but not with phytotoxicity. Bajwa and Weston [67] identified chlorogenic acid and ambrosin as compounds positively correlated with germination inhibition, while vatirenene was found in high concentrations (19.8%) in the roots. Vatirenene was also detected in breath tests, reported by Koo andThomas [68], potentially linked to aspergillosis. Additionally, caryophyllene oxide demonstrates effective fungicidal properties [69] and can induce resistance in reproductive parts [70, 71], while globulol exhibits strong antibacterial activity [72]. Our current investigation highlights the flower extract of *P. hysterophorus* as a rich source of chemical compounds responsible for antioxidant and insecticidal activities.

While various studies have explored the phytochemistery, pharmacology, and other biological activities of *P. hysterophorus*, there is limited research on its antioxidant activity, insecticidal properties against *B. brassicae*, and chemical analysis of the flower extract. Given the

growing interest in botanical-based natural antioxidants for the food, medical, and agrochemical industries, this study fills a critical research gap by investigating the antioxidant and insecticidal potential of *P. hysterophorus* flower extract against *B. brassicae*, providing valuable insights into its novel applications.

## Conclusions

*Parthenium hysterophorus*, despite being a noxious weed, exhibits significant beneficial properties for the medical and agrochemical industries. The flower extract displays remarkable antioxidant and insecticidal activities, particularly against *Brevicoryne brassicae*, a widespread sucking pest. The extract's potency is further supported by its $LC_{50}$ values of 1446μgmL$^{-1}$ and 750μgmL$^{-1}$ at 72 and 96 hours, respectively. The strong correlation between antioxidant activity, phenolic/flavonoid content, and extract yield highlights its potential. Chemical analysis reveals the presence of various bioactive compounds responsible for these activities. Therefore, *P. hysterophorus* extract shows promise as an alternative to synthetic chemicals for antioxidant and insecticidal purposes. However, further research is necessary to optimize extraction, purification, and evaluate the biological potential of these compounds.

## Acknowledgments

The support and supervision provided by Professor Zumin Gu, Ji Mingshan and experimental guidelines of lab mates of Biopesticides Laboratory, Plant Protection College Shenyang is greatly acknowledged.

## Author Contributions

**Data curation:** Zumin Gu.

**Funding acquisition:** Mingshan Ji, Zumin Gu.

**Investigation:** Mingshan Ji, Zumin Gu.

**Methodology:** Maqsood Ahmed, Ansar Javeed, Aatika Sikandar.

**Software:** Xuejing Bai.

**Supervision:** Zumin Gu.

**Validation:** Mingshan Ji.

**Visualization:** Zumin Gu.

**Writing – original draft:** Maqsood Ahmed.

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
