## [Decision Letter · Decision Letter 0]

14 Aug 2022

PONE-D-22-14804Antioxidant, Insecticidal Activity and Chemical Profiling of Flower’s Extract of Carrot Grass (Parthenium hysterophorus L.)PLOS ONE

Dear Dr. Gu,

Thank you for submitting your manuscript to PLOS ONE. After careful consideration, we feel that it has merit but does not fully meet PLOS ONE’s publication criteria as it currently stands. Therefore, we invite you to submit a revised version of the manuscript that addresses the points raised during the review process.

We look forward to receiving your revised manuscript.

Kind regards,

Ali Bajwa, Ph.D

Academic Editor

PLOS ONE

Journal Requirements:

2. In the 'header' sections of your manuscript the following text is present 'Nextgenediting Template PLOS_One_Word_Template.docx. www.nextgenediting.com [double click header to delete]'. Did you receive any third party support in conducting this research, analyzing the data, or preparing the manuscript for submission? If yes, provide details as to the organization(s) involved and their specific contributions by updating the acknowledgements section of the manuscript.

“G.Z.;

Grant No. (0201300)

National Key Research & Development Program of China

No;”

Additional Editor Comments:

The manuscript reports on bioactivity of parthenium weed flower extracts with some significant results. However, authors must address comments raised by reviewers, especially Reviewer 2. I don't think Carrot grass is a good common name to use when in fact the weed is a broadleaf species.

Reviewers' comments:

Reviewer's Responses to Questions

**Comments to the Author**

1. Is the manuscript technically sound, and do the data support the conclusions?

Reviewer #1: Yes

Reviewer #2: Partly

2. Has the statistical analysis been performed appropriately and rigorously? 

Reviewer #1: Yes

Reviewer #2: I Don't Know

3. Have the authors made all data underlying the findings in their manuscript fully available?

Reviewer #1: Yes

Reviewer #2: Yes

4. Is the manuscript presented in an intelligible fashion and written in standard English?

Reviewer #1: Yes

Reviewer #2: No

5. Review Comments to the Author

Reviewer #1: It is a good piece of work. However, needs some corrections before publication.

1- Do not use useless abbreviations. Please see comments in abstract.

2- Whole paper needs thorough corrections for formatting. Give proper spacing between a digit and a its units.

3- Introduction is too lengthy. Delete the first paragraph.

4- Delete Table 1.

5- Check stats in Table 2 for flavonoids.

6- In Table 5, give units of RT

7- Format references uniformly and correctly.

8- Two or more references should be cited at the end of Paragraph 2 of Introduction. I have suggested two references. Please check their suitabllity.

Reviewer #2: This is a good study but the English language/expression is very poor and it needs to be improved. For specific comments please see the attached file. The Figure 1 does not include the positive control treatment, please check.

6. PLOS authors have the option to publish the peer review history of their article (what does this mean?). If published, this will include your full peer review and any attached files.

Reviewer #1: **Yes: **Dr. Arshad Javaid

Reviewer #2: **Yes: **Asad Shabbir

---

## [Author Response · Author response to Decision Letter 0]

15 Jan 2023

Reviewer Comments:

Reviewer 1

Review comments:

Page 9

Point 1: innate of what? do you mean native?

Response: 

The word innate changed to inherent 

Point 2.3; need to explain how?

Response: 

Sentence was explained in detail to clarify the meaning

Point 4,5 No clear, check the grammar

Response: compliance has been done and sentence was revised

Point 6, 7 Again, this need to be explained how?

Response compliance has been done and explained to clear the meaning 

Point 8, 9 Is this correct referencing style?

Response: reference style has been adjusted according the journal

Point, 10 not clear

Response sentence was revised to clarify the meaning

Page 10

Point 1,2,3 Do you mean parthenin?

Response: yes, compliance was done 

Point 4,5,6 move 'health' before 'consequences'

Response: compliance has been done and sentence was revised as suggested 

Point 7,8 Do you mean "come in contact'

Response: yes the same meaning

Point 8, add author citations with the sci. names of all plants when mentioned first time

Response: compliance has been made and instruction were followed

Page 16

Point 1 highest?

Response: compliance was done 

Page 21 

Point 1 leaves.? 

Response: sentence was revised.

Point 2 Either use Parthenium or Parthenium hysterophorus, throughout

Response: compliance has been done and instruction was followed.

Page 36

Point C.L stands for?

Response: actually the word is F.L stands for Fiducial Limit

Page 39

Point 1G = +ve control missing in Figure

Response: Positive control was added in the figure

Reviewer 1

Point 1 Also add authority

Response: compliance was made and Authority was added 

Point 2 Do not use such useless abbreviations. Write full words for these names or parameters.

Response: Instructions were followed and useless abbreviation were deleted 

Point 3 Give space

Response: compliance was made

Point 4 showed.?

Response: compliance was made and the word changed to “showed”

Point: 5 No need of First paragraph.

Response: First paragraph deleted.

Point 6 CIte reference correctly. Delete the name of authors

Response: reference was cited in correct style

Point 7 This paragraph should be extended

Response: The suggested paragraph was extended.

Point 8 It is ......

Response Compliance has been made

Point 9 Add a references here.; Following latest references is suggested:

Response The suggested reference were added and cited in the manuscript 

Point 10 Should not be a new paragraph

Response Compliance was made and Instructions were followed

Point: 11 No need to mention the Institute.

Response: Information about institute were deleted

Point: 12 Do not use capital letters 

Response: suggestion were followed, and word were corrected 

Point: 13 Delete %

Response: The symbol % was removed

Point: 14 Add space 

Response: The space was added where needed

Point 15 Do not use upper case letters for the names of compounds

Response:: Compliance has been done 

Point 16 No need of this introductory paragraph

Response: A part of the paragraph has been deleted

Point 17 Format references uniformly and correctly 

Response: References were formatted according to the journal style

---

## [Editor Report · Decision Letter 1]

14 Feb 2023

PONE-D-22-14804R1Antioxidant, Insecticidal Activity and Chemical Profiling of Flower’s Extract of Carrot Grass (Parthenium hysterophorus L.)PLOS ONE

Dear Dr. Gu,

Thank you for submitting your manuscript to PLOS ONE. After careful consideration, we feel that it has merit but does not fully meet PLOS ONE’s publication criteria as it currently stands. Therefore, we invite you to submit a revised version of the manuscript that addresses the points raised during the review process.

We look forward to receiving your revised manuscript.

Kind regards,

Aishat Akere, Ph.D

Staff

PLOS ONE

Journal Requirements:

Additional Editor Comments (if provided):

**Manuscript is acceptable as the authors have addressed the reviewers' comments. I suggest authors replace the common name 'carrot grass' with 'parthenium weed' in title and throughout the manuscript at proofs stage.**
---

## [Author Response · Author response to Decision Letter 1]

28 Feb 2023

Rebuttal Letter

Title: 

[Antioxidant, Insecticidal Activity and Chemical Profiling of Flower’s Extract of Parthenium weed (Parthenium hysterophorus L.]

[PlosOne]

[Submission Date: May 26, 2022]

Dear Editor 

Thank you for inviting us to submit a revised draft of our manuscript entitled, "Antioxidant, Insecticidal Activity and Chemical Profiling of Flower’s Extract of Parthenium weed (Parthenium hysterophorus L. PlosOne journal. We also appreciate the time and effort you and each of the reviewers have dedicated to providing insightful feedback on ways to strengthen our paper. Thus, it is with great pleasure that we resubmit our article for further consideration. 

We have incorporated changes that reflect the detailed suggestions you have graciously provided. We also hope that our edits and the responses we provide below satisfactorily address all the issues and concern you and the reviewers have noted.

To facilitate your review of our revisions, the following is a point-by-point response to the questions and comments.

EDITOR COMMENTS [1st round]

The present manuscript is informative and interpretation of obtained results is satisfactory. But the presentation of the manuscript is poor and must be improved significantly as suggested by the reviewers. Moreover, detail method used for the Extraction and Purification of Biochemical Compound should be described in little detail or relavant reference should be provided. 

Negative and positive control must be considered rationally in each assay as one of the major concerns of the reviewer. 

Authors must be careful in presenting the units [unit of LC50 of residual assay in table 3 is µg/mL while in text (result section 2.3 Insecticidal Activity) mentioned mg/mL]. 

Response: Compliance has been done; 

Manuscript has edited by native English speaker/TOPEDIT SCIENTIFIC EDITING [Http://www.topeditsci.com]

Negative and Positive control has been added where needed as suggested by the reviewers.

In Table 3 is µg/mL has been corrected as point out by the reviewer.

Reviewer Comments:

Reviewer 1

1. Introduction:

1- Introduction part should be started by specific paragraph regarding the microbial resistance, infectious diseases, reactive species, oxidative stress, and the role of naturally occurring compounds in the treatment of such issues. 

Instructions have been followed and a paragraph has been added at the start of the introduction as suggested by the reviewer along with references; 

“Oxidative stress plays a double role in infections; the pathologies that arise during such infections can be attributed to oxidative trauma and the creation of reactive species, often with lethal consequences. Microbial resistance to conventional antibiotics poses a significant threat to the treatment of infectious diseases. However, phytochemicals exhibit latent biological activity towards both resistant and sensitive pathogens. Phytochemicals are a valuable source of bioactive compounds with antimicrobial activities. Among these phytochemicals, phenolics are diverse secondary metabolites such as tannins, flavonoids, and lignin that exhibit antioxidant properties and are abundant in plant tissues. Likewise, reactive oxygen species (ROS) are produced as typical products in plant cellular breakdown. Naturally occurring compounds play a vital role against microbial resistance in the management of infectious diseases.”

2- The sentence ''Natural plants are venerated source of….'' should be typed as ''Medicinal plants are venerated source of….''.

Response: 2. Compliance has been done and stated as Medicinal plants are venerated source of phytochemical compounds….….….

3. Discussions:

1- The sentence ''Natural plants are God gifted treasures…..'' should be typed as ''Medicinal plants are God gifted treasures…..''.

Response: 1. Instructions have been followed and statement has been revised as Medicinal plants are God gifted treasures for humans which possess …..

2- Please, re-write this section ''Moreover, all of the populations of C. colocynthis extract showed antibacterial activity against Pseudomonas aeruginosa and Escherichia coli, Enterococcus faecalis and Staphylococcus aureus and antifungal activity against four Candidia species i.e. Candida krusei, Candida glabrata, Candida parapsilosis and Candida albicans 32'', try to delete the repeated word (and). 

Response: 2. Instructions have been followed and statement has been revised as suggested 

Moreover, all of the populations of C. colocynthis extract showed antibacterial activity against Pseudomonas aeruginosa, Escherichia coli, Enterococcus faecalis and Staphylococcus aureus whereas, antifungal activity against four Candidia species i.e. Candida krusei, Candida glabrata, Candida parapsilosis and Candida albicans 32

3- In the section ''Similarly, a flavonoid4’-methoxy-5,7-dihydroxyflavone6-C-glucoside isolated.'', the term (-C-) should be typed in italic font. 

Response: 3. Instructions have been followed and the term has been revised as

Similarly, a flavonoid 4’-methoxy-5,7-dihydroxyflavone 6-C-glucoside isolated 

Materials and Methods:

1- If allowed, a molecular docking study should be performed.

Response: 1. As labs are closed, shortage of time and mainly, COVID-19 Pandemic and Omicron etc. the docking study at this stage may not be possible; however in future this should be followed in another experiment 

Abbreviations: 

1- A list of abbreviations should be inserted by the end of the manuscript before references.

Response: 1. Compliance has been done as suggested by the reviewer 

1. DPPH (1, 1-diphenyl-2-picrylhydrazyl) 

2. EC50 (Half maximal effective concentration)

3. LC50; (Lethal concentration)

4. ABTS (2, 2'-Azino-Bis-3-Ethylbenzothiazoline-6-Sulfonic Acid)

5. TEV (Tobacco Etch Virus )

6. CMV (Cucumber Mosaic Virus)

7. Conc. (Concentration)

8. PDA (Potatoes Dextrose Agar)

9. WP (Wettable Powder)

References: 

1. All Scientific names of plants should be typed in italic fonts.

Response: 1. All the scientific names of the pants has been changed into italic fonts in the ref list

2. All Scientific names of microorganisms should be typed in italic fonts.

Response: 2. All the scientific names of microorganisms has been changed into italic fonts 

3. In some references; page number, volume number, and issue number were missed.

Response: 3. Compliance has been done; page number, volume number, and issue number has been added as suggested.

4. Ref. No. 10: The word (linn.) should be typed as (Linn.), also the plant name should be typed as (Luffa acutangula). 

Response: 4. Instructions have been followed and the word (linn.) typed as (Linn.), as well as re-typed as Luffa acutangula.

Ref No 10; Soam, P. S., Singh, T. & Vijayvergia, R. Short Communication Citrullus colocynthis (Linn.) and Luffa acutangula (l.) Roxb, schrad. Source of bioinsecticides and their contribution in managing climate change. in (2013).

5. Ref. No. 10: The word (And) should be typed as (and). 

Response: 5. Instructions have been followed, And written as “and” 

Ref No 10; Soam, P. S., Singh, T. & Vijayvergia, R. Short Communication Citrullus colocynthis (Linn.) and Luffa acutangula (l.) Roxb, schrad. Source of bioinsecticides and their contribution in managing climate change. in (2013).

6. In some references; the complete list of authors were missed except first author (e.g., Ref. No. 12; Ahmed, M. et al.). 

Response: 6. Instructions have been followed and a complete list of authors has been added in the references list.

Supplementary Files S1:

1. Add the word ''and'' before the last 1H-NMR and 13C-NMR value and the chemical shift unit (ppm).

Response: 6. Compliance has been done as suggested like……………… 

1D (1H-NMR and 13C-NMR),

REVIEWER 2

1. There are numerous typos and grammatical mistakes that I’m not able to point them one by one (for example, using Upper-Lower case for the name of compounds, disease and pathogens; repeated full name of pathogen; not using comma before "respectively", and so on)

Response: 1. Compliance has been done and the Manuscript also has been revised and edited by a English speaker/ TOPEDIT SCIENTIFIC EDITING [Http://www.topeditsci.com]

2. Spinasterol and 22,23-dihydrospinasterol presented as mixture. But the author does not give any evidence for a 5:4 ratio of this mixture. The MS and NMR spectra were not clear to determine the ratio of this mixture compounds.

Response: 2. The mass spectrometer examination showed molecular ion peak (M) at m/z 414 and 412 which suggested the molecular formula as a mixture of C29H50O, C29H48O with the ratio of 5:4 calculated by the available results of mass spectrum; elemental analysis and nuclear magnetic resonance analysis (1H-NMR and 13C-NMR) suggested the ratio of 5:4.

3. These authors did not show the positive and negative control to indicate the activities of these compounds (Table 1, 3, 4, 5). So the data is not enough to conclude the biological activities of this mixture.

Response: 3. Compliance has been done and positive as well as negative control has been added in the table where needed as;

Table 1. For DPPH assay methanol was used as a negative control and the DPPH solution only (without the sample). Whereas, DPPH+ methanol was used as experimental control. Therefore, negative control was usually methanol and DPPH. Negative control was prepared to obtain the absorbance of the DPPH before reacting with the sample. 

However, the positive control was prepared in order to compare the results of the samples with those obtained from a standard compound. 

Table 4. Positive control was added in the Table 3 which was missed by mistake, 

Table 3. As LC50 was calculated from the raw data obtained during calculation of mortality, So, it was not compulsory to calculate or mention the mortality of positive control. However, positive control was mentioned now in the Table 4.

Table 5. In the Table 5 Correlation of antioxidant activity versus antifungal and insecticidal activity was presented, as data was used from the already prepared tables, so, only correlation was calculated.

2nd round 

ADDITIONAL EDITOR SUGGESTIONS: [2nd round]

1. [Manuscript is acceptable as the authors addressed the reviewer’s comments. I suggest the author replace the common name ‘carrot grass’ with ‘parthenium weed’ in title and throughout the manuscript at proof stage.]

RESPONSE: [The common name ‘carrot grass’ has been replaced with ‘parthenium weed’ in title and throughout the manuscript]

REVIEWER 1 COMMENTS:

1[Innate of what? do you mean native?]

 RESPONSE: [The word innate changed to inherent] 

2[Point 2.3; need to explain how?]

 RESPONSE: [Sentence was explained in detail to clarify the meaning]

3[Point 4,5 No clear, check the grammar]

RESPONSE: [compliance has been done and sentence was revised]

4[Point 6, 7 Again, this need to be explained how?]

 RESPONSE: [compliance has been done and explained to clear the meaning]

5[Point 8, 9 Is this correct referencing style?]

RESPONSE: [reference style has been adjusted according the journal]

6 [Point, 10 not clear]

RESPONSE: [sentence was revised to clarify the meaning]

7[Point 1,2,3 Do you mean parthenin?]

RESPONSE [yes, compliance was done]

8[Point 4,5,6 move 'health' before 'consequences'?]

RESPONSE: [compliance has been done and sentence was revised as suggested]

9[Point 7,8 Do you mean "come in contact']

 RESPONSE: [: yes the same meaning]

10 [Point 8, add author citations with the sci. names of all plants when mentioned first time

 RESPONSE: [compliance has been made and instruction were followed]

11 [Point 1 highest?

 RESPONSE: [compliance was done]

12 [Point 1 leaves.?]

 RESPONSE: [: sentence was revised.]

13 [Point 2 Either use Parthenium or Parthenium hysterophorus, throughout

 RESPONSE: compliance has been done and instruction was followed.

14 [Point C.L stands for?]

 RESPONSE: actually the word is F.L stands for Fiducial Limit

15 [Point 1G = +ve control missing in Figure]

 RESPONSE: Positive control was added in the figure

Reviewer 2

1 [Point 1 Also add authority]

 RESPONSE: compliance was made and Authority was added 

2 [Point 2 Do not use such useless abbreviations. Write full words for these names or parameters.]

 RESPONSE: Instructions were followed and useless abbreviation were deleted 

3 [Point 3 Give space]

 RESPONSE: compliance was made

4 [Point 4 showed.?]

 RESPONSE: compliance was made and the word changed to “showed”

5 [Point: 5 No need of First paragraph.]

 RESPONSE: First paragraph deleted.

6 [Point 6 CIte reference correctly. Delete the name of authors]

 RESPONSE: reference was cited in correct style

7 [Point 7 This paragraph should be extended]

 RESPONSE: The suggested paragraph was extended.

8 [Point 8 It is ......]

 RESPONSE Compliance has been made

9 [Point 9 Add a references here.; Following latest references is suggested:]

 RESPONSE The suggested reference were added and cited in the manuscript 

10 [Point 10 Should not be a new paragraph]

 RESPONSE Compliance was made and Instructions were followed

11 [Point: 11 No need to mention the Institute.]

 RESPONSE: Information about institute were deleted

12 [Point: 12 Do not use capital letters] 

 RESPONSE: suggestion were followed, and word were corrected 

13 [Point: 13 Delete %]

 RESPONSE: The symbol % was removed

14 [Point: 14 Add space] 

 RESPONSE: The space was added where needed

15 [Point 15 Do not use upper case letters for the names of compounds]

 RESPONSE: Compliance has been done 

16 [Point 16 No need of this introductory paragraph]

 RESPONSE: A part of the paragraph has been deleted

17 [Point 17 Format references uniformly and correctly] 

 RESPONSE: References were formatted according to the journal style 

CONCLUDING REMARKS: Again, thank you for giving us the opportunity to strengthen our manuscript with your valuable comments and queries. We have worked hard to incorporate your feedback and hope that these revisions persuade you to accept our submission.

Sincerely,

Zumin Gu.

Corresponding Author

Associate Professor, College of Plant Protection

Shenyang Agricultural University, Shenyang

No.120 Dongling Road Shenhe District Liaoning, CHINA

guzumin1212@163.com

Tel: +86-24- 8848‐7148; Fax: +86-24-8834-2

---

## [Decision Letter · Decision Letter 2]

11 Apr 2023

PONE-D-22-14804R2Antioxidant, Insecticidal Activity and Chemical Profiling of Flower’s Extract of Carrot Grass (Parthenium hysterophorus L.)PLOS ONE

Dear Dr. Gu,

Thank you for submitting your manuscript to PLOS ONE. After careful consideration, we feel that it has merit but does not fully meet PLOS ONE’s publication criteria as it currently stands. Therefore, we invite you to submit a revised version of the manuscript that addresses the points raised during the review process.

We look forward to receiving your revised manuscript.

Kind regards,

Samuel Adelani Babarinde, PhD

Academic Editor

PLOS ONE

Reviewers' comments:

Reviewer's Responses to Questions

**Comments to the Author**

1. If the authors have adequately addressed your comments raised in a previous round of review and you feel that this manuscript is now acceptable for publication, you may indicate that here to bypass the “Comments to the Author” section, enter your conflict of interest statement in the “Confidential to Editor” section, and submit your "Accept" recommendation.

Reviewer #1: (No Response)

Reviewer #3: All comments have been addressed

2. Is the manuscript technically sound, and do the data support the conclusions?

Reviewer #1: Partly

Reviewer #3: Yes

3. Has the statistical analysis been performed appropriately and rigorously? 

Reviewer #1: Yes

Reviewer #3: Yes

4. Have the authors made all data underlying the findings in their manuscript fully available?

Reviewer #1: Yes

Reviewer #3: Yes

5. Is the manuscript presented in an intelligible fashion and written in standard English?

Reviewer #1: No

Reviewer #3: Yes

6. Review Comments to the Author

Reviewer #1: The revision seems incomplete.

1- Significant improvement in language and formatting is needed throughout the paper. Better to get it edit by a professional language editor.

2- Rewrite Abstract in a better scientific language.

3- Delete 1st paragraph of Introduction.

4- Improve Discussion.

5- Format references uniformly and correctly.

6- Also add latest references.

Reviewer #3: The authors have addressed all the comments raised by reviewers satisfactorily, hence can be accepted in its present form.

7. PLOS authors have the option to publish the peer review history of their article (what does this mean?). If published, this will include your full peer review and any attached files.

Reviewer #1: **Yes: **Prof. Dr. Arshad Javaid

Reviewer #3: **Yes: **Shailendra S Gurav

---

## [Author Response · Author response to Decision Letter 2]

26 May 2023

A file is uploaded ad rebuttal letter as well as Response to reviewers

---

## [Decision Letter · Decision Letter 3]

29 Aug 2023

PONE-D-22-14804R3Antioxidant, Insecticidal Activity and Chemical Profiling of Flower’s Extract of Carrot Grass (Parthenium hysterophorus L.)PLOS ONE

Dear Dr. Gu,

Thank you for submitting your manuscript to PLOS ONE. After careful consideration, we feel that it has merit but does not fully meet PLOS ONE’s publication criteria as it currently stands. Therefore, we invite you to submit a revised version of the manuscript that addresses the points raised during the review process.

We look forward to receiving your revised manuscript.

Kind regards,

Samuel Adelani Babarinde, PhD

Academic Editor

PLOS ONE

Journal Requirements:

Reviewers' comments:

Reviewer's Responses to Questions

**Comments to the Author**

1. If the authors have adequately addressed your comments raised in a previous round of review and you feel that this manuscript is now acceptable for publication, you may indicate that here to bypass the “Comments to the Author” section, enter your conflict of interest statement in the “Confidential to Editor” section, and submit your "Accept" recommendation.

Reviewer #1: (No Response)

Reviewer #3: All comments have been addressed

2. Is the manuscript technically sound, and do the data support the conclusions?

Reviewer #1: Yes

Reviewer #3: Yes

3. Has the statistical analysis been performed appropriately and rigorously? 

Reviewer #1: Yes

Reviewer #3: Yes

4. Have the authors made all data underlying the findings in their manuscript fully available?

Reviewer #1: Yes

Reviewer #3: Yes

5. Is the manuscript presented in an intelligible fashion and written in standard English?

Reviewer #1: No

Reviewer #3: Yes

6. Review Comments to the Author

Reviewer #1: Paper needs a thorough revision. It is full of formatting errors. Please especially concentrate on the following points:

1- There should be proper spacing between a digit and its unit.

2- write units in a similar format throughout the manuscript.

3- Write scientific names in Italics. It is a big issue in the references.

4- Write equations correctly, in same font size and style.

5- Write P≤0.05 instead of P>0.05.

6- Format references uniformly and correctly. Titles of papers should be in sentence format.

Reviewer #3: All queries raised by respective reviewers have been resolved and revised manuscript can be accepted in the present form.

7. PLOS authors have the option to publish the peer review history of their article (what does this mean?). If published, this will include your full peer review and any attached files.

Reviewer #1: **Yes: **

Reviewer #3: No

---

## [Author Response · Author response to Decision Letter 3]

12 Oct 2023

Dear Editor 

Now this version is final with compliance to all the comments suggested by the editors and reviewers. Now plz accept it for publication. 

Regards

---

## [Decision Letter · Decision Letter 4]

11 Dec 2023

Antioxidant, Insecticidal Activity and Chemical Profiling of Flower’s Extract of Carrot Grass (Parthenium hysterophorus L.)

PONE-D-22-14804R4

Dear Dr. Gu,

We’re pleased to inform you that your manuscript has been judged scientifically suitable for publication and will be formally accepted for publication once it meets all outstanding technical requirements.

Kind regards,

Samuel Adelani Babarinde, PhD

Academic Editor

PLOS ONE

Reviewers' comments:

Reviewer's Responses to Questions

**Comments to the Author**

1. If the authors have adequately addressed your comments raised in a previous round of review and you feel that this manuscript is now acceptable for publication, you may indicate that here to bypass the “Comments to the Author” section, enter your conflict of interest statement in the “Confidential to Editor” section, and submit your "Accept" recommendation.

Reviewer #1: All comments have been addressed

2. Is the manuscript technically sound, and do the data support the conclusions?

Reviewer #1: Yes

3. Has the statistical analysis been performed appropriately and rigorously? 

Reviewer #1: Yes

4. Have the authors made all data underlying the findings in their manuscript fully available?

Reviewer #1: Yes

5. Is the manuscript presented in an intelligible fashion and written in standard English?

Reviewer #1: Yes

6. Review Comments to the Author

Reviewer #1: This is the fourth round of evaluation. Authors have incorporated all the suggested corrections. Paper is now acceptable.

7. PLOS authors have the option to publish the peer review history of their article (what does this mean?). If published, this will include your full peer review and any attached files.

Reviewer #1: **Yes: **Prof. Dr. Arshad Javaid

---

## [Editor Report · Acceptance letter]

15 May 2024

PONE-D-22-14804R4 

PLOS ONE

Dear Dr. Gu, 

I'm pleased to inform you that your manuscript has been deemed suitable for publication in PLOS ONE. Congratulations! Your manuscript is now being handed over to our production team.

Kind regards, 

on behalf of

Dr. Samuel Adelani Babarinde 

Academic Editor

PLOS ONE